



# The 'urban meteorology island': a multi-model ensemble analysis

Jan Karlický[1,2], Peter Huszár[1], Tereza Nováková[1], Michal Belda[1], Filip Švábik[1], Jana Ďoubalová[1,3], and Tomáš Halenka[1]

[1]Department of Atmospheric Physics, Faculty of Mathematics and Physics, Charles University, Prague, V Holešovičkách 2, 180 00 Prague 8, Czech Republic
[2]Institute of Meteorology and Climatology, Department of Water, Atmosphere and Environment, University of Natural Resources and Life Sciences, Vienna, Gregor-Mendel-Straße 33, 1180 Vienna, Austria
[3]Czech Hydrometeorological Institute (CHMI), Na Šabatce 17, 14306, Prague 4, Czech Republic

**Correspondence:** J. Karlický (Jan.Karlicky@mff.cuni.cz)

**Abstract.**

Cities and urban areas are well-known for their impact on meteorological variables and thereby modification of the local climate. Our study aims to generalize the urban-induced changes of specific meteorological variables by introducing a single phenomenon – the urban meteorology island (UMI). A wide ensemble of 24 model simulations with the WRF and RegCM regional climate models on European domain was performed to investigate various urban-induced modifications as individual components of the UMI. The results show that such an approach is meaningful, because nearly in all meteorological variables considered, statistically significant changes occur in cities. Besides previously documented urban-induced changes of temperature, wind speed and boundary-layer height, the study is focused also on changes of cloud cover, precipitation and humidity. An increase of cloud cover in cities, together with a higher amount of sub-grid scale precipitation is detected in summer afternoons. Specific humidity is significantly lower in cities. Further, the study shows that different models and parameterizations can have a strong impact on discussed components of UMI. Multi-layer urban scheme with anthropogenic heat considered increases winter temperatures by more than 2 °C and reduces wind speed more strongly than other urban models. Also the selection of planetary boundary-layer scheme influences the urban wind speed reduction, as well as boundary-layer height with the greatest extent. Finally, urban changes in cloud cover and precipitation are mostly sensitive to the parameterization of convection.

## 1 Introduction

Climate is one of the most important factors that influences the conditions for life at a specific place. Considering the fact that half of current population lives in cities (United Nations, Department of Economic and Social Affairs, 2014) and the population of largest cities is still projected to increase (Baklanov et al., 2016), and that the total number of cities of different sizes is rising (Mirzaei, 2015), the general knowledge of typical urban climate features and the difference with respect to their rural counterparts is becoming more and more crucial.

The most well-known urban climate feature is the so-called urban heat island (UHI), firstly described several decades ago (Oke and Maxwell, 1975), which means, in simple terms, that the urban temperatures are higher compared to rural ones. In following years, a large number of UHI observations was performed (e.g. Oke, 1982; Godowitch et al., 1985; Wolters and





Brandsma, 2012; Theeuwes et al., 2015) and several empirical relation for UHI intensity (difference of city centre and vicinity

temperature) were introduced (Oke, 1982; Theeuwes et al., 2017). More recently, once the computational power enabled a finer grid resolution, many of modelling studies, focusing primarily on UHI were performed (e.g. Ryu et al., 2013; Huszar et al., 2014; Trusilova et al., 2016; Göndöcs et al., 2017; Karlický et al., 2018; Huang et al., 2019). Specialized models of urban canopies within standard numerical weather prediction and regional climate models were used to capture the specifics of urban climate features.

However, several observation and model-based studies show that other meteorological variables are significantly altered by urban canopies too. The impact of cities on boundary-layer structure was documented already many years ago (Godowitch et al., 1985; Oke, 1987; Angevine et al., 2003), similarly to the impact on wind flow (Oke, 1987; Klein et al., 2001; Droste et al., 2018). These modifications of atmospheric dynamics over cities have considerable consequences to mixing, dispersion of pollutants and air quality in urban areas, which is also confirmed by model studies investigating the urban-induced changes

of dispersion conditions (Karlický et al., 2018) and of primary or secondary pollutant concentrations (Fallmann et al., 2016; Huszár et al., 2018; Huszar et al., 2018; Li et al., 2019; Huszar et al., 2020; Ďoubalová et al., 2020).

Recently, some studies have investigated the impact of urban canopies on convection, cloud cover and rainfall. E.g. Theeuwes et al. (2019) describe the observed cloud cover enhancement over Paris and London during summer as a consequence of increased convection caused by UHI. Manola et al. (2020) show increased summer precipitation intensities and also overall

precipitation increase in the city of Amsterdam in comparison to rural surroundings, also as a result of enhanced convection over the urban area. Enhanced convection, turbulence and mixing can cause, under special weather conditions, even wind speed increase over cities (Droste et al., 2018). Finally, humidity is also impacted by urban canopy, e.g. Langendijk et al. (2019) show a significant decrease of relative humidity in Berlin (primarily in summer) and a less expressed decrease in specific humidity.

All these urban-induced meteorological changes, forming island-like features in the spatial distribution of the mentioned

variables over urban areas, resulted in a formulation of new concepts describing the urban weather and climate. Apart from the UHI (Oke and Maxwell, 1975), other meteorology-related "urban islands" were defined recently: urban dry island (UDI; Moriwaki et al., 2013), urban cool island (UCI; Theeuwes et al., 2015), surface urban heat island (SUHI; Göndöcs et al., 2017) and urban wind island (UWI; Droste et al., 2018). Such a relatively wide range of well documented island-like perturbations of physical fields over urban areas in contrast to their rural counterparts basically means that the meteorological conditions differ

due to changes of the whole atmospheric physics and dynamics in urban areas. Therefore, we introduce here a generalization of the above mentioned urban area islands: the urban meteorology island (UMI) phenomenon, which we consider as an urban area that has significantly different meteorological conditions from surrounding rural areas. The specific "one-variable" islands like UHI, UDI, UCI, SUHI and UWI can be regarded as components of UMI, denoted as UXI, in general.

Considering the above-mentioned modelling studies, models well simulate, at least qualitatively, most of the processes

leading to the UHI and other elements of the UMI are also simulated with all expected features. However, large differences exist in the magnitude of individual elements of the UMI between different models and their configurations (e.g. Trusilova et al., 2016; Karlický et al., 2018; Huang et al., 2019; Huszar et al., 2020). It is clear that the resulting UMI will be highly dependent on how the relevant UXI-forming processes are represented in models, which include parameterization of sub-grid





processes as boundary-layer turbulence, convection, air-surface heat and water exchange, micro-physics and urban canopy
physics. Also the choice of driving model itself can have a potentially strong impact.

Motivated by this, here, we present a novel study that 1) perceives the urban-induced meteorological changes as one UMI
concept where urban perturbations of specific variables (UXI) are considered as components of the whole UMI. Further, 2) a
large number of cities from different locations in a model domain is taken into account to enable a robust, city-independent
view on the impact of a specific model and parametrization on the resulting UMI. Next, 3) a wide range of the model simulation
ensemble provides a robust estimation of different components of UMI, including their magnitudes and temporal evolution.
Finally, 4) a multi-variable validation of the whole model ensemble is performed here that brings an useful view for other
model users and can be feedback for model developers.

The paper is composed as follows: after the Introduction, the models, the data and the design of the experiments is presented,
followed by the Results section that contains a detailed validation of different model set-ups and the comparison of UMI
components. Finally, the results are discussed and conclusions are drawn.

## 2    Models and data

### 2.1    Models used

To achieve results of the study, several model simulations were performed by involving two meteorological models, namely
the Weather Research and Forecasting (WRF; Skamarock et al., 2008) model in version 4.0.3 and Regional Climate Model
(RegCM; Giorgi et al., 2012) in version 4.7. In both cases, the ERA-interim data (Dee et al., 2011) were used as a driving
meteorology. The computational domain was also the same for both models, specifically a $190\times166$ domain with 9 km hor-
izontal resolution over Europe, centred over Prague, Czech Republic (Fig. 1, Fig. 2). The simulation time range is 2 years
(2015–2016), with December 2014 as spin-up. Static geographic data are taken from standard WRF and RegCM input, only
land-use fields are derived from CORINE Land Cover data, version CLC 2012. While WRF uses a dominant land-use (i.e.
one land-use type for a particular grid-box, Fig. 1), RegCM works with a fractional land-use (more land-use types included
proportionally in one grid-box, Fig. 2). Urban canopy parameters are the same as in Karlický et al. (2018).

The selection of model schemes is based on their availability, restrictions in their different combinations, expected impact on
the urban effects and to enable cumulus radiation feedback. In all WRF simulations, the radiative transfer is parameterized by
the Rapid Radiative Transfer Model for General Circulation Models (RRTMG; Iacono et al., 2008), microphysical processes
are resolved by the Purdue Lin scheme (Chen and Sun, 2002) and land-surface exchange by the Noah land-surface model (Chen
and Dudhia, 2001). The specific WRF simulations differ in parameterizations of boundary-layer processes, which are resolved
by the Mellor–Yamada–Janjic scheme (MYJ; Janjić, 1994) or the BouLac PBL scheme (Bougeault and Lacarrere, 1989) and
in surface layer processes (SFL) description, using the scheme as in Eta model (Janjić, 1994) or the Revised MM5 scheme
(Jiménez et al., 2012). Further, multi-layer Building Environment Parameterization (BEP; Martilli et al., 2002) linked to the
Building Energy Model (BEM; Salamanca et al., 2009), the Single-Layer Urban Canopy Model (SLUCM; Kusaka et al., 2001)
and the bulk parameterization are tested for processes in urban environment. Convection is parameterized by the Grell-Freitas

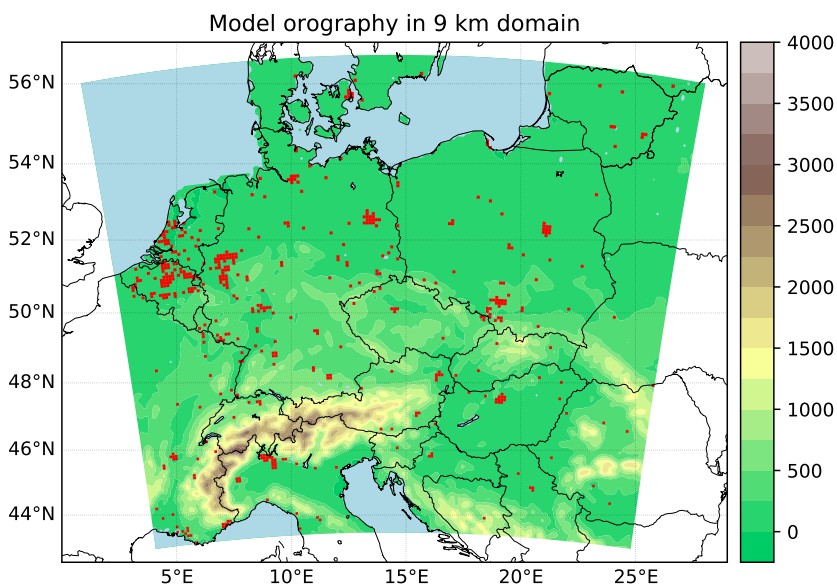

**Figure 1.** Position of model domain with model orography (m) and grid-boxes marked with red, in which the urban land-use type is dominant.

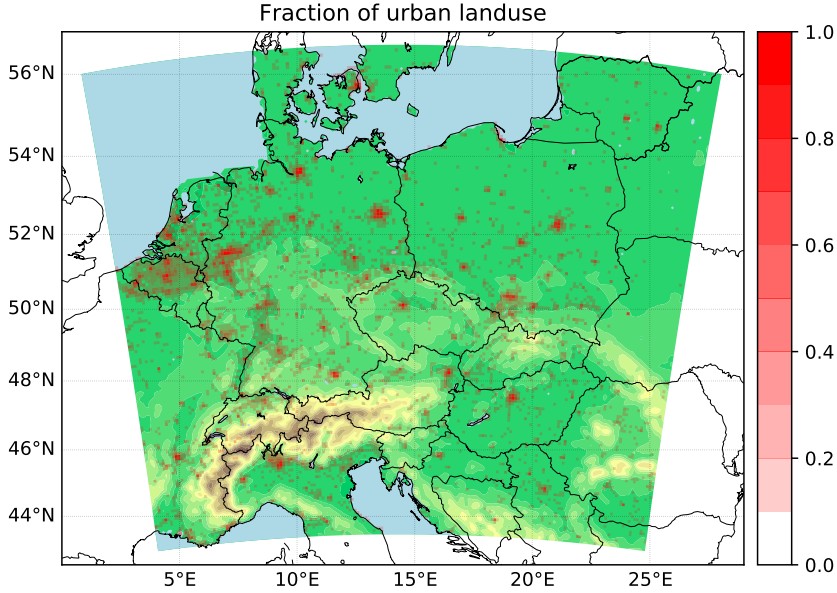

**Figure 2.** As Fig. 1 but with red marked urban land-use proportion.





**Table 1.** WRF model setups.

| Experiment | Urban model | # levels | PBL scheme | SFL scheme | Convection |
|---|---|---|---|---|---|
| WU3L22 | BEP+BEM | 49 | MYJ | Eta | Grell-Freitas |
| WU3L82 | BEP+BEM | 49 | BouLac | Eta | Grell-Freitas |
| WU3L81 | BEP+BEM | 49 | BouLac | MM5 | Grell-Freitas |
| WU1L22 | SLUCM | 40 | MYJ | Eta | Grell-Freitas |
| WU1L22C5 | SLUCM | 40 | MYJ | Eta | Grell-3D |
| WU1L22C1 | SLUCM | 40 | MYJ | Eta | Kain-Fritsch |
| WU1L82 | SLUCM | 40 | BouLac | Eta | Grell-Freitas |
| WU1L82C5 | SLUCM | 40 | BouLac | Eta | Grell-3D |
| WU1L82C1 | SLUCM | 40 | BouLac | Eta | Kain-Fritsch |
| WU1L81 | SLUCM | 40 | BouLac | MM5 | Grell-Freitas |
| WU1L81C5 | SLUCM | 40 | BouLac | MM5 | Grell-3D |
| WU1L81C1 | SLUCM | 40 | BouLac | MM5 | Kain-Fritsch |
| WU0L22 | bulk | 40 | MYJ | Eta | Grell-Freitas |
| WU0L82 | bulk | 40 | BouLac | Eta | Grell-Freitas |
| WU0L81 | bulk | 40 | BouLac | MM5 | Grell-Freitas |

(Grell and Freitas, 2014), Grell-3D (Grell, 1993) and Kain-Fritsch scheme (Kain, 2004). In total, 15 specific combinations of chosen schemes were used and they are described in Table 1.

In terms of RegCM setup, all simulations are run on 40 model layers, the NCAR Community Climate Model Version 3
(CCM3; Kiehl et al., 1996) is used to parameterizing of the radiation transfer, land-surface transfer is resolved by the CLM4.5 (Lawrence et al., 2011; Oleson et al., 2013) scheme and urban-canopy-layer processes are solved by the Community Land Model Urban (CLMU; Oleson et al., 2008) scheme. Here, the model ensemble is created by two different parameterizations of boundary-layer processes – the University of Washington (UW; Grenier and Bretherton, 2001; Bretherton et al., 2004) and the Holtslag (Holtslag et al., 1990) PBL scheme, by two different approaches of solving sub-grid convection – the Tiedtke scheme
(Tiedtke, 1989) and the Grell scheme (Grell, 1993) and by three schemes of microphysical processes (MP) – explicit one-moment scheme by Nogherotto et al. (2016), older SUBBEX scheme (Pal et al., 2000) and the explicit 5-class single moment WSM5 model (Hong et al., 2004). The specific combinations of chosen schemes in terms of RegCM model are described in Table 2, the first model setup is considered as a baseline setup.

## 2.2  Validation data

To assess the model biases for temperature and precipitation over the entire domain, the E-OBS dataset (Haylock et al., 2008) of version 17.0 is chosen. Further, station-based data from European Climate Assessment and Dataset (ECAD; Klein Tank et al., 2002) are used for more detailed validation of model results in terms of daily maximum and minimum temperatures,





**Table 2.** RegCM model setups.

| Experiment | PBL scheme | Convection | MP scheme |
|------------|-----------|-----------|-----------|
| RU | UW | Tiedtke | Nogherotto |
| RUHo | Holtslag | Tiedtke | Nogherotto |
| RUG | UW | Grell | Nogherotto |
| RUS | UW | Tiedtke | SUBBEX |
| RUW | UW | Tiedtke | WSM5 |
| RUHoW | Holtslag | Tiedtke | WSM5 |
| RUGW | UW | Grell | WSM5 |
| RUHoS | Holtslag | Tiedtke | SUBBEX |
| RUGS | UW | Grell | SUBBEX |

**Table 3.** Used station from ECAD for specific variables.

| T2max/T2min | Cloud cover | Humidity | SW radiation | Wind speed |
|-------------|-------------|----------|--------------|------------|
| Dresden-Klotzche | Dresden-Klotzche | Dresden-Klotzche | Dresden-Klotzche | Dresden-Klotzche |
| Wien-Hohe Warte | Wien-Hohe Warte | Veliki Dolenci | Grossenzersdorf | Veliki Dolenci |
| Budapest | Hurbanovo | Novi Sad | Timisoara | Hurbanovo |
| Beograd | Beograd | Beograd | Craiova | Nove Mesto |
| Zagreb-Gric | Zagreb-Gric | Zagreb-Gric | Ostrava-Poruba | Celje |
| Warszawa-Okecie | Kosice | Gorlitz | Belsk | Kosice |
| Berlin-Dahlem | Berlin-Dahlem | Berlin-Dahlem | Potsdam | Berlin-Dahlem |
| Hamburg-Fuhlsbuettel | Hamburg-Fuhlsbuettel | Hamburg-Fuhlsbuettel | Hamburg-Fuhlsbuettel | Hamburg-Fuhlsbuettel |
| Muenchen | Muenchen-Stadt | Muenchen | Nurnberg | Muenchen |
| Falsterbo | Falsterbo | Falsterbo | Schleswig | Falsterbo |

cloud cover, relative humidity, downward shortwave radiation and wind speed. For every variable, ten stations were chosen that are equally-distributed over the domain. The specific stations are listed in Table 3.

Model surface (skin) temperature is validated by data from the Moderate Resolution Imaging Spectroradiometer (MODIS), operated by Terra and Aqua satellites (Wan et al., 2015a, b). For further use, monthly values of surface temperature in 0.05° horizontal resolution are selected. These are computed only from observations with clear-sky conditions, when satellite sensors are able to scan the earth surface. In our area of interest, four observations per day are performed, approximately at 10 and 21 UTC by the Terra satellite and at 2 and 12 UTC by the Aqua satellite.





## 3 Results

### 3.1 Model validation of model simulation ensemble

First, the general comparison of model temperatures and precipitation against E-OBS data is performed. Seasonal temperature and rainfall sums are averaged over the whole domain (excluding twenty rows and columns at the domain edges) and compared with corresponding E-OBS data. Results are shown in Fig. 3. Most of the WRF simulations predict average temperatures correctly, only simulations with MYJ PBL and Eta SFL schemes give notable underestimations (up to 2 °C). RegCM simulations with the Nogherotto MP scheme produce substantial biases, mainly in spring and summer season. Precipitation biases are related mainly to the chosen convection scheme: for WRF, the average summer overestimation is partly reduced by using the Grell-3D scheme. In RegCM, the Nogherotto MP scheme influences precipitation (besides temperature), resulting in a high bias during the whole year. Further, it seems that experiments with the Grell convection scheme are marked with higher summer overestimation than those with the Tiedtke convection.

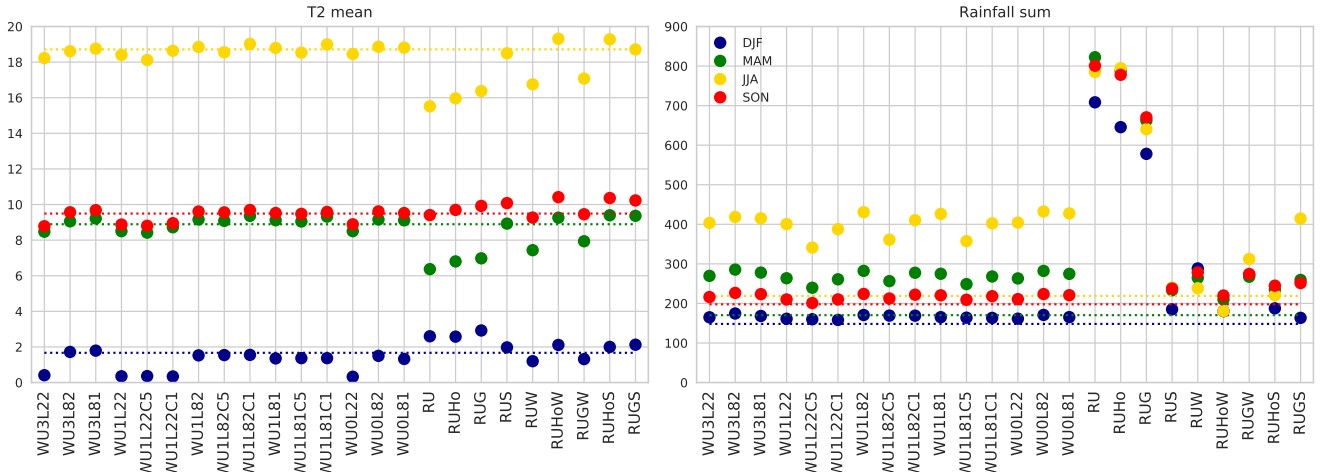

**Figure 3.** Domain averaged seasonal temperatures (°C) and precipitation sums (mm) for individual simulations and E-OBS data (dotted lines).

Daily ECAD values from selected stations are used for more detailed model validation over European urban areas, which are the main focus of the paper. Bias, correlation coefficient and regression coefficient (slope) are computed for all model simulations between ECAD and model time-series including values for every day of specific season and for all stations. The results are presented in Fig. 4 (winter season) and Fig. 5 (summer season).

Because measuring stations are located mainly in cities (Table 3), the type of the urban canopy parameterization impacts the results too. E.g. the BEP+BEM urban canopy model gives temperature extremes clearly higher than SLUCM or bulk and with bias up to 2 °C. Again, the combination of MYJ PBL and Eta SFL schemes gives lower temperatures than other combinations, which mostly means a negative temperature bias. The remaining WRF simulations predict temperatures reasonably well. For



**Figure 4.** Statistical evaluation of individual simulations using ECAD station data for selected variables in winter season. Units are °C (temperature), fractions of one (cloud cover and relative humidity), $W\,m^{-2}$ (downward shortwave radiation) and $m\,s^{-1}$ (wind speed).





**Figure 5.** Same as Fig. 4 but for summer season.



RegCM, simulations with the Nogherotto MP scheme exhibit highest biases in minimum temperatures, as well as in case of
cloud fraction. The biases of humidity and radiation are less than 0.1 and 20 W m$^{-2}$, respectively. Wind speed is overestimated
by models, but the biases vary significantly between specific models and setups; the BEP+BEM urban model and simulations
with the MYJ PBL and Eta SFL schemes, as well as all RegCM simulations, make a less bias. Correlations of temperatures,
radiation and wind speed are, in general, around 0.8 or higher. On the other hand, for cloud fraction and humidity, they are only
about 0.6. However, RegCM simulations with Nogherotto MP have the correlation substantially lower (except wind speed).
In summer, again, RegCM simulations with Nogherotto MP are marked with highest biases and the lowest correlations
(with exception of minimum temperature and wind speed). Further, WRF simulations predict temperatures with biases less
than 2 °C, while biases of some other RegCM simulations are up to 4 °C in case of maximum temperature. Biases of cloud
fraction and humidity are mostly less than 0.1 and the sign of cloud fraction bias is clearly linked to the sign of radiation bias.
WRF simulations are characterized by positive radiation bias up to 50 W m$^{-2}$. In the case of RegCM, a negative radiation bias
prevails. As in winter, simpler urban models produce higher wind speeds in cities that results in higher positive biases. Biases
of RegCM model for wind speed depend mainly on choice of PBL scheme. Correlations between model and observed values
are of similar magnitude as in winter.

Being an important parameter of the urban canopy thermodynamic state, we also compare the modelled surface (skin)
temperatures with satellite-based data. With consideration of the fact that satellite-based data are available only in specific times
of day, roughly at 2 UTC, 12 UTC (Aqua satellite), 21 UTC and 10 UTC (Terra satellite), and the comparison of both daytime
and nighttime satellite data with corresponding model values shows very similar features, averaging over both satellites is
performed to get simple daytime and nighttime satellite data of surface temperature. Seasonal and domain averages of daytime
and nighttime data together with the corresponding model simulation results are displayed in Fig. 6. Considering the fact that
only clear-sky days are included in satellite-based data, Fig. 6 offers a rough comparison of modelled and observed values. In
general, summer surface temperatures are predicted more accurately than winter ones, where model values are significantly
higher (by 2–4 °C). Analogously to temperature biases against E-OBS (Fig. 3), the highest deviations are detected in RegCM
simulations with the Nogherotto MP scheme, mainly in summer daytime (exceeding 5 °C). Again, WRF simulations with the
MYJ PBL and Eta SFL schemes give surface temperatures up to 2 °C lower than remaining WRF simulations. Modelled and
observed surface temperatures are consistent in feature that daytime spring values are higher than autumn ones, while nighttime
spring surface temperatures are lower than autumn ones.

## 3.2   Components of the Urban Meteorology Island

As the most important component of the UMI, we show here the air temperature alteration over urban areas. Because of many
previous studies describing the classical UHI, we show only a multi-model average of temperatures for chosen big cities over
the domain for winter and summer season (Fig. 7). The UHI is clearly visible with a magnitude of about 1 °C in winter and
1.5 °C in summer, even using a relatively coarse model resolution (9 km).

Similarly, surface (skin) temperature is significantly altered by urban surfaces, which is obvious from satellite-based mea-
surements of surface temperature, during clear-sky conditions. The same procedure as in the previous section was used to





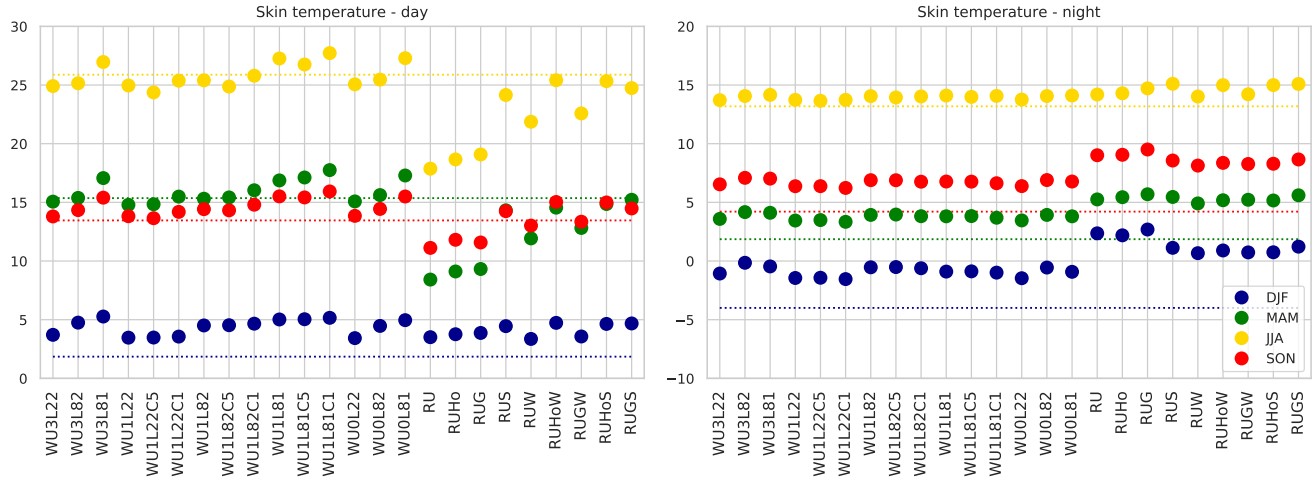

**Figure 6.** Averaged seasonal whole-domain surface temperatures (°C) in daytime and nighttime given by specific simulations and satellites (dotted line).

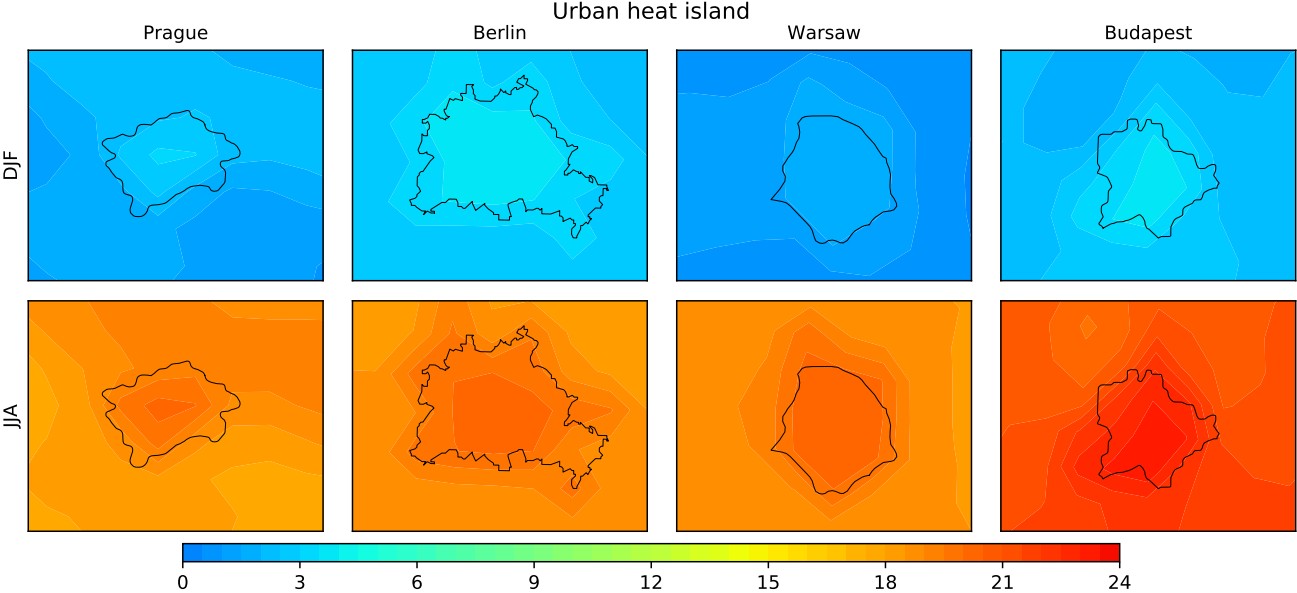

**Figure 7.** Multi-model averaged winter and summer temperatures (°C) around Prague, Berlin, Warsaw and Budapest





determine daytime (Fig. 8) and nighttime (Fig. 9) values. SUHI is most pronounced in summer daytime, when urban temperatures are approximately 4 °C higher. During nighttime, the SUHI intensity is slightly smaller, about 3 °C. In winter, SUHI is
not so clearly pronounced as in summer and reaches 2 °C, except for Warsaw and Budapest where it is very small.

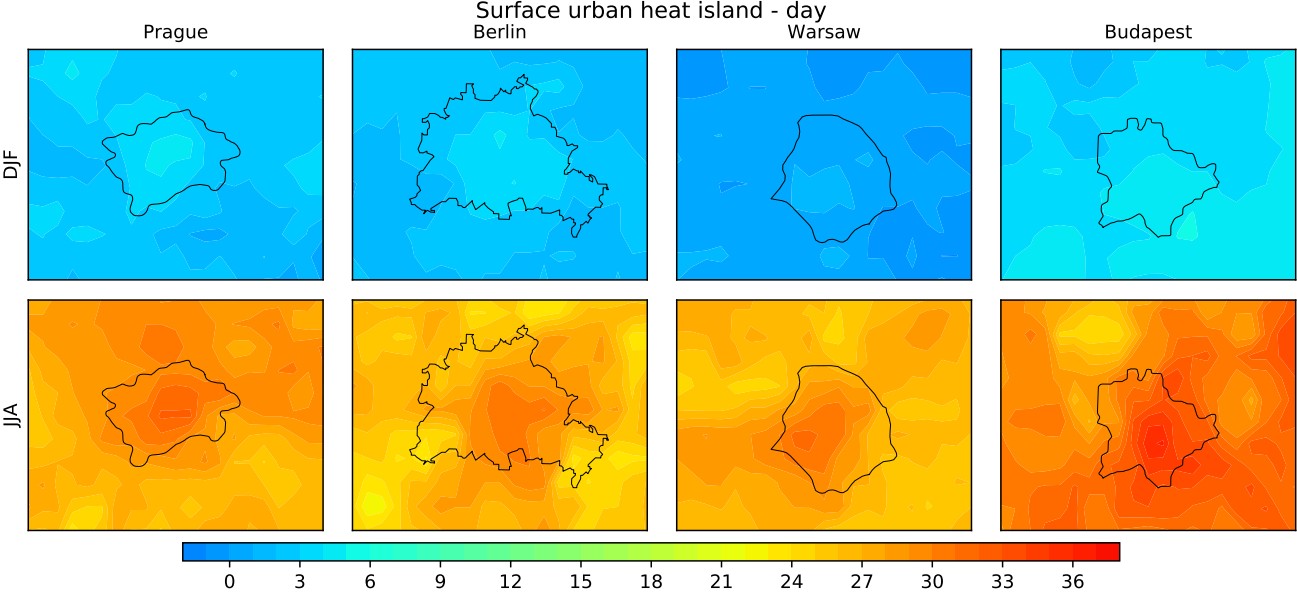

**Figure 8.** Satellite based daily winter and summer surface temperatures (°C) around Prague, Berlin, Warsaw and Budapest.

For further investigation of UMI components and to eliminate the effect of specific conditions occurring in different cities, we have chosen ten large European cities across the whole model domain (Tab. 4). For every city, we define the city centre together with three points in its surroundings in a distance of 30 km from the centre forming an equilateral triangle with one apex directed to the north. The bilinear interpolation is used to determine model values in the chosen locations. Values from
three surroundings points are averaged to get one value for city centre and one value for vicinity for each city.

Firstly, different components of UMI are investigated in their diurnal cycles for every season. One sample t-test on 98 % of significance level was used to determine the statistical significance of non-zero difference between city centre and vicinity values averaged over all models and cities, separately for specific season and hour in day. The results for winter and summer seasons are shown in Fig. 10, with the red color meaning the statistical significance. UMI elements as air and surface tempera-
ture, boundary-layer, moisture flux are always significantly altered by urban surfaces, and with some exceptions also the wind speed, specific humidity and sub-grid scale precipitation. Cloud cover is influenced by cities in specific times of day, most continuously in summer afternoon and evening. The significance test did not reveal cross-model significant impact on wind direction and large-scale precipitation.

Alterations of temperature, wind speed and boundary-layer height in urban areas and their diurnal cycles are described in
detail in many previous studies (e.g. Huszar et al., 2014; Huszár et al., 2018; Karlický et al., 2018; Huszar et al., 2020), therefore





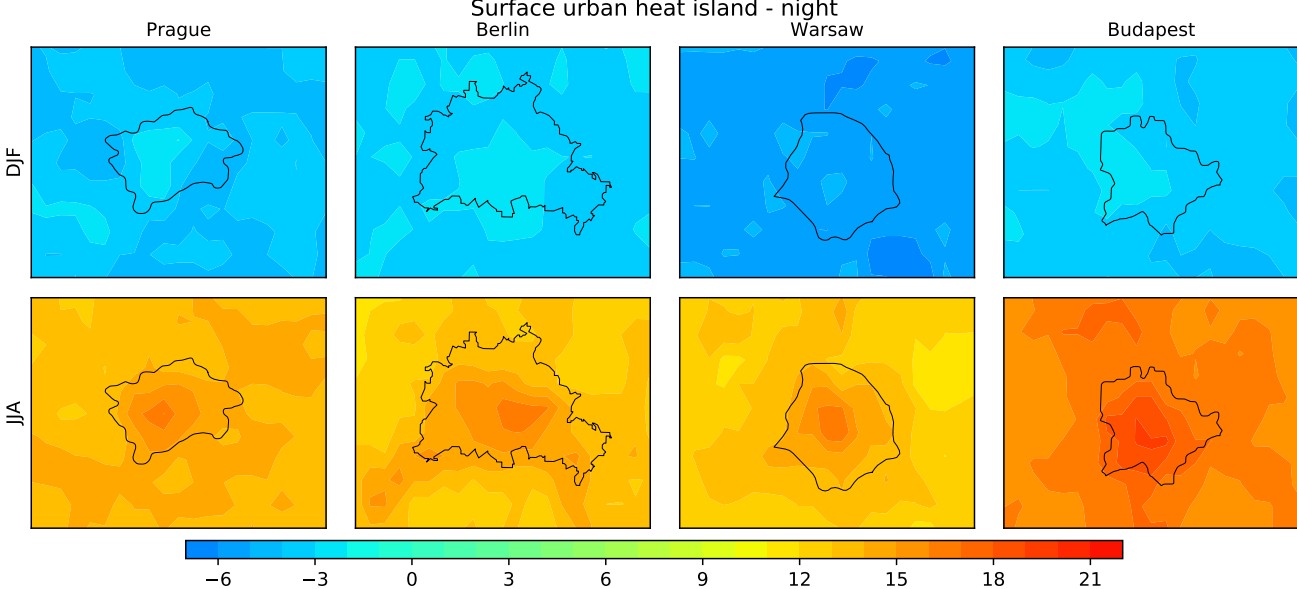

**Figure 9.** Same as Fig. 8 but for nighttime.

**Table 4.** Positions of city centres and vicinities for chosen cities.

|  | city centre | | vic. 1 – 30 km N | | vic. 2 – 30 km SWW | | vic. 3 – 30 km SEE | |
|---|---|---|---|---|---|---|---|---|
|  | lat | lon | lat | lon | lat | lon | lat | lon |
| Prague | 50.075 | 14.44 | 50.345 | 14.44 | 49.94 | 14.076 | 49.94 | 14.804 |
| Vienna | 48.208 | 16.387 | 48.478 | 16.387 | 48.073 | 16.023 | 48.073 | 16.751 |
| Budapest | 47.5 | 19.076 | 47.77 | 19.076 | 47.365 | 18.712 | 47.365 | 19.44 |
| Beograd | 44.811 | 20.461 | 45.081 | 20.461 | 44.676 | 20.097 | 44.676 | 20.825 |
| Zagreb | 45.802 | 15.984 | 46.072 | 15.984 | 45.667 | 15.62 | 45.667 | 16.348 |
| Warsaw | 52.244 | 21.017 | 52.514 | 21.017 | 52.109 | 20.653 | 52.109 | 21.381 |
| Berlin | 52.521 | 13.408 | 52.791 | 13.408 | 52.386 | 13.044 | 52.386 | 13.772 |
| Hamburg | 53.594 | 9.986 | 53.864 | 9.986 | 53.459 | 9.622 | 53.459 | 10.35 |
| Munich | 48.147 | 11.567 | 48.417 | 11.567 | 48.012 | 11.203 | 48.012 | 11.931 |
| Copenhagen | 55.676 | 12.504 | 55.946 | 12.504 | 55.541 | 12.14 | 55.541 | 12.868 |



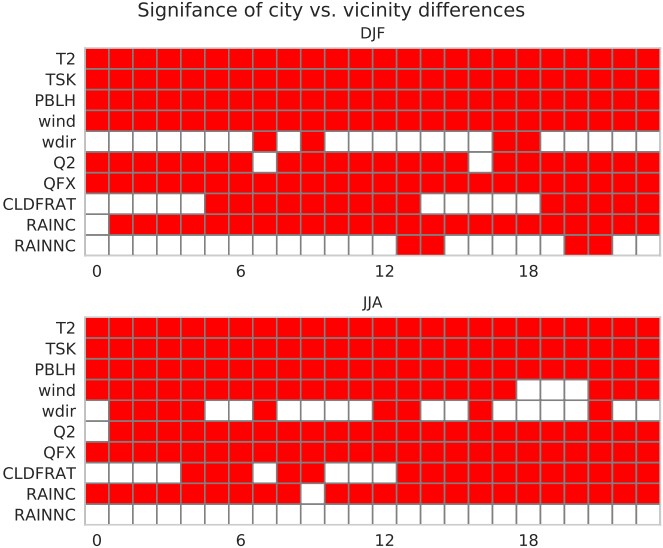

**Figure 10.** Significance of urban meteorology island components during the day. From above: 2-metre air temperature (T2), surface temperature (TSK), boundary-layer height (PBLH), wind speed (wind), wind direction (wdir), specific humidity (Q2), upward moisture flux (QFX), cloud cover (CLDFRAT), sub-grid scale precipitation (RAINC), large-scale (resolved) precipitation (RAINNC). Red color means statistical significance on the 98% level.

other components of the UMI will be discussed in more detail in the following paragraphs. In Fig. 11, differences between urban and rural cloud cover are presented. Statistically significant differences are detected mainly in summer afternoon and during evening, probably as a result of the enhanced convection, as the sub-grid scale precipitation is also increased in cities during this time (Fig. 12). In winter, a small but significant reduction of cloud cover is detected in models from morning to noon and during night.

Fig. 12 shows the alterations of sub-grid scale precipitation. The most distinct feature is the summer afternoon and evening increase, above 5 mm per season during some hours. However, a significant increase of sub-grid scale precipitation is detected nearly in the whole winter and summer diurnal cycle, but with much smaller magnitude: up to 1 mm in remaining summer hours and up to 0.3 mm in the winter diurnal cycle.

In terms of specific humidity, a statistically significant urban decrease prevails during summer and winter daytime (Fig. 13). The magnitude in summer (up to $1 \times 10^{-3}$ $\text{kg}\,\text{kg}^{-1}$) is more pronounced compared to winter (up to $0.2 \times 10^{-3}$ $\text{kg}\,\text{kg}^{-1}$). During nighttime, a slight humidity increases are detected (up to $0.1 \times 10^{-3}$ $\text{kg}\,\text{kg}^{-1}$).

### 3.3 Impact of models and their parameterizations

Here we focus on the analysis of how different parameterization influences the resulting UMI component. Fig. 14 shows a spread of city-vicinity differences for different components of UMI, separately for every model simulation. In general,





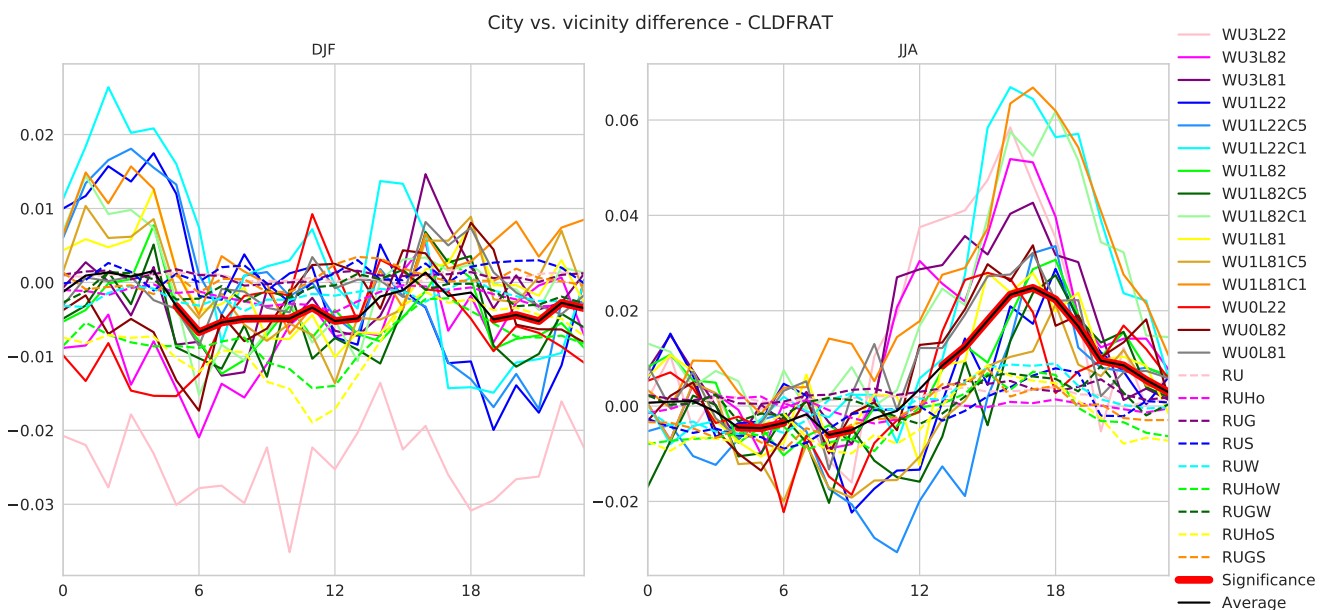

**Figure 11.** Diurnal cycle of city-vicinity differences in cloud cover (in fractions of one).

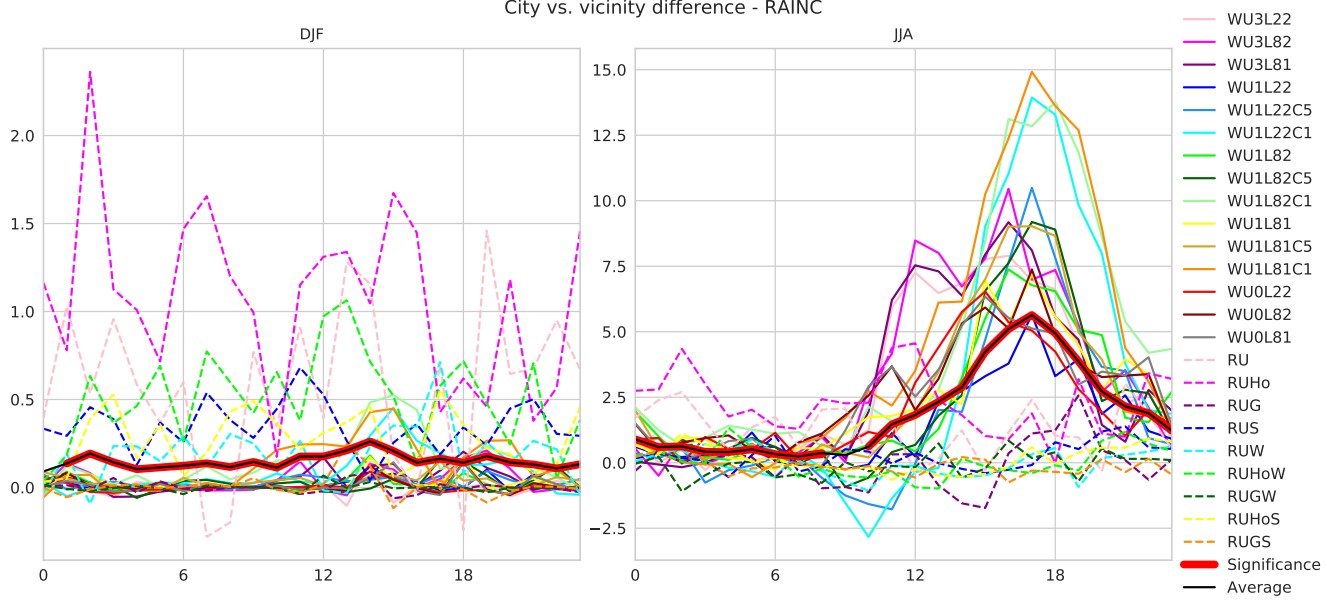

**Figure 12.** Diurnal cycle of city-vicinity differences for sub-grid scale precipitation (in mm, differences between seasonal sums).



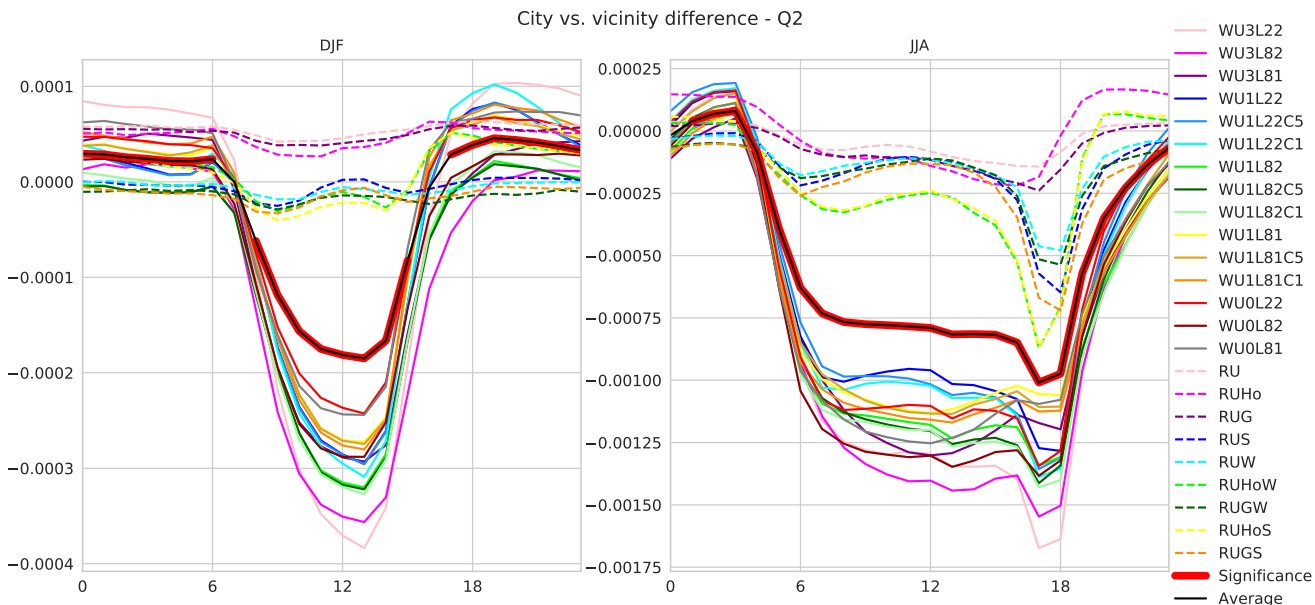

**Figure 13.** Diurnal cycle of city-vicinity differences for specific humidity (in $kg\,kg^{-1}$).

differences between WRF and RegCM results are large. In case of city-vicinity differences of seasonal daily temperature, in WRF simulations they are about 2 °C in summer and about 1–1.5 °C in winter, while simulations with the BEP+BEM urban parameterization make the difference even larger, up to 3 °C. On the other hand, in RegCM simulations, differences are much lower – about 1 °C only. Similarly, in terms of specific humidity, city-vicinity differences in summer are also more pronounced

in WRF simulations (0.6–1.0×$10^{-3}$ $kg\,kg^{-1}$), while RegCM gives differences up to 0.3×$10^{-3}$ $kg\,kg^{-1}$. In winter, the urban specific humidity reduction reaches 0.1×$10^{-3}$ $kg\,kg^{-1}$.

Also the enhancement of PBL height in urban centers is greater in summer than in winter. In this season, the positive change is mainly between 200 and 300 metres, but in terms of RegCM model, only the simulations with the Holtslag PBL scheme without Nogherotto MP reach such values. In winter, the difference is smaller: over cities, PBL is about 100 m higher than

over vicinities and only the BEP+BEP urban model makes this difference higher. In terms of wind speed, the highest urban reductions are detected (independently on season) in WRF simulations with the BEP+BEP urban scheme. Simulations with the SLUCM urban scheme give the smallest reduction, but it also depends on the PBL parameterization. Combination of MYJ PBL and Eta SFL schemes makes the wind speed reduction about 1 $m\,s^{-1}$, combination of BouLac and Eta schemes produces the reduction of about 0.5 $m\,s^{-1}$ and the combination of BouLac and MM5 schemes results in even a slight wind

speed increase. In terms of WRF simulations with the bulk urban parameterization, it also depends on PBL and SFL schemes: in winter a reduction occurs but in summer the increase dominates. RegCM simulations exhibit urban wind speed reductions up to 1 $m\,s^{-1}$ and they are higher in winter than in summer.



The impact of cities on cloud cover in summer is mostly positive, with a high dependency on model simulation. In winter, a cloud cover reduction dominates. In terms of sub-grid scale precipitation, WRF simulations produce significant summer increase in cities (mostly between 50 and 100 mm), but the winter urban modifications are negligible. In most RegCM simulations, the precipitation increase is much smaller in comparison to WRF simulations.

## 4 Discussion

The temperature underestimation of WRF simulations with the MYJ PBL and Eta SFL schemes (Fig. 3) in winter season are similar as detected by Karlický et al. (2018), summer temperatures are probably influenced by Tiedtke convection scheme (Tiedtke, 1989), which is not tested in this study, but making smaller summer overestimation of precipitation, related to higher summer temperatures. Also Zhong et al. (2017), who used the MYJ scheme of PBL in their study, show a slight temperature underestimation. BouLac PBL scheme seems to give more accurate temperature means and extremes. In terms of RegCM simulations, great temperature biases in simulations with the Nogherotto MP scheme imply a great overprediction of cloud cover, specific humidity and underestimation of downward shortwave radiation both in winter (Fig. 4) and summer (Fig. 5). This leads also to great precipitation biases and a slight temperature overprediction in winter. The remaining RegCM simulations show smaller temperature biases in summer, even less than Huszár et al. (2018), who used a very similar model setup. Huszar et al. (2020) used the model configuration corresponding to RU simulation (i.e. the Nogherotto MP, Tiedtke convection and UW PBL schemes), but it gives, in 9 km resolution, smaller biases: summer underestimation and winter overestimation of temperatures in range of 1–2 °C and precipitation overprediction by approximately 100 mm.

Focusing on cities, differences caused by distinct urban schemes appear. Winter positive bias of temperature in BEP+BEM simulations are well in-line with our previous study (Karlický et al., 2018). Huang et al. (2019) also described lower wind speeds and thereby more reduced bias in simulation with BEP+BEM in comparison to SLUCM. On the contrary, correlation of model and observed data of wind speed with values of the coefficient between 0.25 and 0.35 is substantially worse than in this study. In terms of temperatures, correlations are comparable.

Validation of model surface temperatures using MODIS satellite data (Fig. 6) results in similar features as the air temperature validation (Fig. 3), keeping in mind that MODIS monthly means are calculated only from days with clear-sky-condition. RegCM simulations with the Nogherotto MP scheme, by overestimating cloud cover during all seasons, show large negative biases (mainly in summer daytime). During nighttime, the positive cloud cover bias leads to temperature increase, mainly in winter. A general agreement of model surface temperatures with values derived from MODIS is shown also by Zhong et al. (2017) in annual averages.

Despite significant differences between specific simulations in terms of the UHI magnitude (Fig. 14), multi-model averaged UHI is still clearly visible around the selected big cities (Fig. 7). It is important to note that no adjustment on the same altitude was made so the results can be influenced partly by orography, which concerns mainly Prague and Budapest, due the relatively complicated terrain within and in the surroundings of these cities. The UHI magnitudes about 1 °C in winter and 1.5 °C in summer are in general agreement with Trusilova et al. (2016), who analysed the observation-based UHI for Berlin. However,





**Figure 14.** Winter (blue) and summer (orange) differences between city and vicinity values of 2-metre air temperature (T2; in °C), specific humidity (Q2; $10^{-3}$ kg kg$^{-1}$), boundary-layer height (PBLH; m), wind speed (wind; m s$^{-1}$), cloud cover (CLDFRAT; fractions of one) and sub-grid scale precipitation (RAINC; mm in seasonal sums). Whiskers indicate 5th and 95th percentiles.





it depends on the location of stations: urban site 'Alexanderplatz' corresponds well with modelled summer UHI, but semi-urban sites 'Tempelhof' and 'Tegel' make the UHI less intensive. Langendijk et al. (2019), presenting results of a wide model ensemble, gives the annual UHI intensity for Berlin in 2015 almost 2 °C. Our previous study (Karlický et al., 2018), presenting observation-based data besides model results, gives smaller UHI intensities in Prague around 0.5 to 1 °C.

The satellite-based observations of surface temperature (Fig. 8 and Fig. 9) confirm the fact that the surface temperatures are significantly affected by urban surfaces too (Fig. 10). The intensities of SUHI around Budapest (Fig. 8 and Fig. 9) are less than given by Göndöcs et al. (2017), who also investigated daily and nocturnal SUHI by MODIS around this city and reached for the city centre and one-week period in summer, 6 °C and 4 °C for daytime and nighttime, respectively. Zhong et al. (2017), who presented an annual mean of surface temperatures taken from MODIS for Yangtze River Delta region of

China, give the intensity of SUHI above 2 °C, which is comparable to our results, if we consider daily and nocturnal values in winter and summer. Satellites sampled the surface temperature only during clear-sky conditions, which means that the SUHI deduced from these measurements is somewhat stronger than the average for the whole season, given the fact that SUHI is more pronounced during sunny days with higher solar input.

    Significance of urban-induced alterations of urban canopy air temperature, surface temperature, PBL height is well docu-
mented in many previous simulation and observation-based studies (e.g. Huszar et al., 2014; Trusilova et al., 2016; Göndöcs et al., 2017; Karlický et al., 2018; Huszár et al., 2018; Huszar et al., 2018, 2020). Also the moisture flux decrease in cities is a well-known phenomenon as shown already by Oke (1987), or more recently by Theeuwes et al. (2019) from latent heat flux comparison. In terms of wind speed changes in urban areas, no statistically significant differences occur during summer evening hours, when the cross-model average is nearly zero. This is probably caused by the combination of UHI, enhanced
turbulence, convection and mixing in a deeper urban boundary-layer, as described by Droste et al. (2018), who concluded that under special conditions and during certain daytimes, the urban wind speed can be even higher.

    Impact of cities on the cloud fraction (Fig. 11) is well documented also by Theeuwes et al. (2019), who investigated cloud cover and its differences in Paris and London and their surroundings during the warm season. They also found a cloud cover urban increase during afternoon and evening, despite drier atmosphere above cities that leads to a cloud base located higher
by approximately 250 m. Higher temperatures in cities probably result in partial dissolution of non-precipitation stratiform clouds and fog in winter, with statistically significant cloud cover reduction during morning hours, because urban precipitation is not reduced (but rather increased, Fig. 12). The reduction of low-level clouds and fog was also found by Yan et al. (2020). In summer, higher temperatures lead to enhanced convection and more frequent occurrence of convective clouds, which leads to increase of sub-grid scale precipitation.

The summer afternoon and evening sub-grid scale precipitation enhancement in cities is also documented by Manola et al. (2020), who analysed observed precipitation features in Amsterdam and its surroundings. Differences occur in summer morning, when they found an increase of the same magnitude as in afternoon. Secondly, they found that the impact of cities on precipitation is higher in winter in relative numbers, which contrasts to our results (except one RegCM experiment). It can be partly explained by choice of cities for investigation (Table 4), that are located in central Europe with greater distance to





oceans and, thus, less maritime climate compared to Amsterdam. Zhu et al. (2019), who analysed the impact of urban areas on precipitation in the Beijing area, also found its positive, but only in the time range of 10–21 hour, which is close to our results.

In terms of impact of cities on specific humidity, Langendijk et al. (2019) gives for the annual difference between Berlin and its surroundings about $0.7 \times 10^{-3}$ kg kg$^{-1}$, which corresponds to our summer values (Fig. 13) and rather to WRF simulations (Fig. 14). Winter differences are much less, even in WRF simulations. During nighttime, a slight humidity increase is visible

(Fig. 13), caused probably by reduced occurrence of dew in cities connected to smaller humidity losses by condensation on surfaces, given that latent heat in cities is still positive during nights, in contrast to surrounding areas (Theeuwes et al., 2019). The daytime urban humidity decrease can be explained by less water availability, enhanced convection and higher vertical turbulent mixing (higher PBL) over cities.

The differences between WRF and RegCM models in terms of intensities of specific UMI elements are largely caused

by the fact that WRF uses dominant land-use (Fig. 1), while RegCM model considers fractional land-use (Fig. 2), so the urban effects in RegCM simulations do not have to be so intensive in city centres and, on the other hand, they are non-zero in their surroundings, where non-zero urban fractions occur, which leads to smaller differences between cities and and their rural vicinities. In other words, the modelled UMI is smoother. This feature is responsible for less UHI intensities in RegCM simulation (Fig. 14). In terms of WRF, higher winter urban temperatures in simulations with the BEP+BEM urban scheme are

consistent with results of previous studies (Liao et al., 2014; Karlický et al., 2018) and are probably caused by higher amounts of anthropogenic heat, internally computed within the scheme. In simulations with the bulk urban scheme, no anthropogenic heat is considered and therefore the winter UHI is suppressed.

As expected, differences between urban and rural PBL heights are sensitive, besides on model and urban scheme, also on the chosen parameterization of boundary-layer processes. In WRF simulations, the MYJ PBL scheme produces a smaller urban

PBL increase compared to the BouLac PBL scheme. Halenka et al. (2019), also using WRF and RegCM to investigating urban effects over Prague, reported urban PBL increases only about 50 m in winter and 100 m in summer in terms of WRF and about 100 m in terms of RegCM (using the Holtslag PBL scheme without Nogherotto MP scheme). In terms of the urbanization induced wind speed changes, again, parameterization of urban processes and PBL are the main influencing factors. Also Halenka et al. (2019) gave the highest wind speed reduction in BEP+BEM simulation, in range 1.5–2 m s$^{-1}$, but in our case of

more cities (Fig. 14), the reduction is much higher. In terms of SLUCM simulations, the reduction is similar, about 1 m s$^{-1}$, similarly in RegCM simulations (about 0.5 m s$^{-1}$ reduction). The wind speed urban increase in bulk simulation is visible only in the summer season in Halenka et al. (2019). The wind speed reduction is mostly less in summer, when higher urban roughness can be compensated by enhanced turbulence, convection and mixing (Droste et al., 2018), i.e. many counteracting effects play a simultaneous role and each model configuration prefers only a subset of these effects.

The impact of cities on cloud cover is clearly influenced by the convection parameterization: Kain-Fritsch scheme makes the summer urban cloud cover increase stronger, despite the fact that this setup does not give the highest precipitation in general. This is very similar to the summer urban increase of sub-grid scale precipitation. The great increases in RegCM simulations with the Nogherotto MP scheme can be explained by overall precipitation bias, where increased urban convection leads to higher difference. In all simulations, it seems that urban changes of sub-grid scale precipitation are largely city-dependent,





in case of RegCM this concerns even the sign of the change. It indicates that some other climate elements such as seasonal total precipitation climatology may influence the urban increase of sub-grid scale precipitation and more research has to be conducted in this regard.

## 5  Conclusions

The study presented results of a model ensemble of 24 simulations using WRF and RegCM models performed over a European
domain covering a two year period. Such a great ensemble enables the robust investigation of the impact of urban surfaces on overall climate and weather in cities, because model uncertainties given by specific model setups were eliminated in the whole ensemble. Urban-induced changes, manifesting in various meteorological variables, were generalised as one concept called urban meteorology island (UMI), where urban perturbations of different meteorological quantities were regarded as components of the overall UMI.
The results of the study showed that this concept is justified and our approach is meaningful, because almost all investigated meteorological variables are significantly altered in cities with respect to their rural counterparts, independently to the chosen city and the model setup. However, quantitative or even (to some extent) qualitative differences were detected between models.

The main conclusions of the presented study are:

- Validation showed large differences between individual model simulations given by different parameterizations of the
different physical processes driving urban meteorology. In RegCM, the microphysics parameterization had the greatest impact on temperature and precipitation biases. The impact of other parameterizations is smaller; Tiedtke convective scheme overpredicts precipitation less than Grell scheme and the Holtslag PBL scheme slightly improves temperature. In WRF, temperature biases are smaller, especially in simulations with BouLac PBL scheme, and significant overestimation is encountered mainly for summer precipitation.

- For ECAD stations (often in cities), the type of urban canopy schemes turned out to be the most important factor determining the model's accuracy. The BEP+BEM urban scheme gives winter temperature minima about 2 °C higher than observed values and reduces positive wind speed bias by 0.5–1 m s$^{-1}$. Most of the model simulations correlate well with ECAD observations, with the correlation coefficient about 0.8 or higher, with exception of cloud cover and relative humidity in winter.

- In general, the UMI components are more pronounced in WRF which used dominant land-use compared to RegCM fractional land-use approach, pointing out the importance of the land-use model representation.

- An increase of cloud cover in cities is modelled, mainly for summer afternoons. This is connected with a summer afternoon urban increase of sub-grid scale precipitation (by 5 mm). These changes are probably caused by an enhanced convection over urban areas, given by higher near surface temperatures while specific humidity is significantly lower in
cities.



- Impact of urban and other parameterization: the BEP+BEM urban model increases the winter UHI by more than 2 °C, and leads to highest reductions of urban wind speeds. In terms of PBL height and its urban increase in summer, the MYJ PBL scheme in WRF simulations makes this UMI element about 50 m smaller, while Holtslag PBL scheme in RegCM increases it to the WRF level. The choice of PBL scheme further influences the wind speed reduction in cities. Modifications of cloud cover and sub-grid scale precipitation in cities are influenced mainly by the parameterization of convection and here, the Kain-Fritsch scheme results in the most pronounced UMI, mainly during summer season.

- Besides results based on model simulations, satellite measurements of surface temperature show significant increase in urban areas too, with the magnitude 3–4 °C in summer and about 2 °C in winter, during clear-sky days.

Our study showed the great importance of multi-model approach when describing the urban meteorological phenomenon, as large differences exist between models in their way of resolving different city and regional scale physical processes. Future research on urban atmospheric processes should thus be based on model/physical ensembles rather than a single model experiments.

*Code and data availability.* Source codes of the WRF and RegCM models are publicly available. Resulting data from simulation performed (stored in the Charles University data storage facilities making about 30 TB) can be obtained upon request to authors.

*Author contributions.* PH designed the main scientific idea and organised the project team, in collaboration with TH. MB prepared the input and validation data; JK and PH performed the model simulation; JK analysed the resulting data, with a contribution of TN, FŠ and JĎ. JK wrote the text and PH revised it.

*Competing interests.* The authors declare that they have no conflict of interest.

*Acknowledgements.* This work has been funded by the Czech Science Foundation (GACR) project No. 19-10747Y and partly by the project OP-PPR (Operation Program Prague – Pole of Growth) CZ.07.1.02/0.0/0.0/16_040/0000383 "URBI PRAGENSI – Urbanization of weather forecast, air quality prediction and climate scenarios for Prague" and by projects PROGRES Q47 and SVV 2020 – Programmes of Charles University. This work was also supported by The Ministry of Education, Youth and Sports from the Large Infrastructures for Research, Experimental Development and Innovations project "IT4Innovations National Supercomputing Center – LM2015070". We further acknowledge the E-OBS dataset from the EU-FP6 project UERRA (http://www.uerra.eu), the Copernicus Climate Change Service, the data providers in the ECA&D project (https://www.ecad.eu) and NASA's Land Processes Distributed Active Archive Center for providing of MODIS satellite data.





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
