# Peer review of "The 'urban meteorology island': a multi-model ensemble analysis"

_Atmospheric Chemistry and Physics, 2020_

## Referee Comment (RC1) · Anonymous Referee #1 · 7 Aug 2020

Review of the manuscript ACP-2020-433:

The 'urban meteorology island': a multi-model ensemble analysis By Jan KarlickÃi, Peter Huszár, Tereza Nováková, Michal Belda, Filip Švábik, Jana Doubalová, and Tomáš Halenka

Remarks:

1. Abstract: the abstract should mention that you only study European cities. In terms of morphology they are substantially different North American cities that this caveat makes sense to mention.

2. Abstract: you indicate that there is a substantial sensitivity of the model to the selected PBL schemes and urban canopy scheme. It would be more attractive for the

reader if you can add a recommendation which settings are preferred.

3. Ln 12: choice => selection

4. Ln 13: hyphenation: boundary-layer scheme

5. Ln 31: hyphenation: boundary-layer structure

6. Ln 59: hyphenation: boundary-layer turbulence. Please check the manuscript throughout.

7. Ln 76: model resolution of 9 km: defend why this is sufficient to represent cities sufficiently. The resolved scales will be 5 \* 9 km = 45 km, which means only cities of that scale are appropriately resolved, but there are not that many cities in Europe of that scale. I the feeling the cities on Belgium and Hungary are larger in your fig 1 than in reality.

8. Ln 78: 2015–2016: please defend why these years have been selected. 2015 is a rather warm year in Europe, so how representative is the selected period.

9. Ln 85: please add some sentences that defend why you have selected these schemes. I expect you have not selected them randomly but that there was a certain strategy or you built upon earlier studies.

10. Table 1: elaborate the table caption. The caption should be placed above the table. Idem for Table 2 and 4.

11. Table 1: please add a sentence that elaborates on the experiment abbreviations. E.g. the "E1U1L82C5" is not naturally related to the <SLUCM, 40, BouLac, Eta, Grell-3D> experiment. All experiment abbreviations start with "E1" so E1 can be removed. Idem for Table 2

12. Ln 118: simulations with MYJ PBL and Eta SFL schemes give notable underestimations (up to 2 deg C): link to literature, this MYJ behaviour is well known.
13. Figure 3: I have reservations against figure 3. Not about the contents but about the plot type. Now the results are shown as time series or at least the lines connect the different experiments. However there are no links between the connected experiments. So the results should be presented differently, e.g. as bar graphs. Idem for Fig 6.

14. Ln 124: Daily ECAD values from selected stations are used for more detailed model validation over european urban areas, which are the main focus of the paper: European should be capitalized.

15. Figure 4: please label all figure a,b,c, etc. This is much more easy for referencing

16. Figure 4: panel T2max and SWDOWN should have an y axis that is better adjusted (less wide range) to the data.

17. Figure 4: in columns 2 and 3 it is very difficult to see what are the differences between the runs. Better to start the y axis at a much higher value. Idem for Fig 5.

18. Figure 4 and 5: if these statistics are averaged over all cities in Table 3, it remains unclear how they are influenced by certain sites or not.

19. Fig 8 and 9: the header "surface heat island" is misleading since this is not plotted according to the caption. Furthermore I do not understand what is the functionality of these plots if you only show satellite data.

20. Ln 198: please be more precise here: about which temperature are writing here. It is daily mean T2m?

21. Ln 202: absolute humidity is the density of the vapour pressure so unit should be g/m3

22. Ln 333: it is unclear whether the statement of BEP+BEM is an advertisement for this scheme or not. Is this scheme the best, despite the biases you report about?

23. Ln 349: I have the feeling the authors are somehow too positive about the satellite data. As far as I understand them, they can only be applied for cloud free days, and

**ACPD**
this does not occur very often, so they may give a biased picture. Please comment.

---

## Referee Comment (RC2) · Anonymous Referee #3 · 15 Sep 2020

This paper deals with urban-induced changes of specific meteorological variables and as such the authors introduce the urban meteorology island (UMI). A large ensemble of 24 model simulations with the WRF and RegCM regional climate models on European domain was performed to investigate various urban-induced modifications as individual components of the UMI. Overall this is an interesting paper with many results which mostly confirm already known aspects of urban areas. The following points need further clarification before paper can be accepted: 1. The fact that the results are achieved using a horizontal resolution of 9 km within the models needs to be mentioned more clearly in the abstract and conclusions. 2. The authors should also motivate why they have chosen the various combinations of physical schemes and possibly list some of the main physical characteristics of the respective schemes to improve readability.

3. Regarding the presentation of the urban results in section 3.2, a selection of those models who performed best in the overall evaluation would make much more sense. 4. What aspects of the study have been surprising for the authors? 5. Can the authors also give a recommendation for the combination of physical packages using either WRF or RegCM?

---

## Author Comment (AC1) · 7 Oct 2020

We would like to thank to Anonymous Referee 3 for all comments, suggestions and corrections in his review of our manuscript. We addressed all and below our point-by-point responses follow:

**Referee's Comment #1:** The fact that the results are achieved using a horizontal resolution of 9 km within the models needs to be mentioned more clearly in the abstract and conclusions.

**Author's response:** We agree with the reviewer, information about the horizontal resolution will be added to the abstract and conclusion. This is although a relatively coarse resolution, but the cities examined cover usually one or more model grid-boxes com-

pletely, moreover in the RegCM model, fractional urban land-use is considered which enables to include the effect of even minor urban areas.

**Referee's Comment #2:** The authors should also motivate why they have chosen the various combinations of physical schemes and possibly list some of the main physical characteristics of the respective schemes to improve readability.

**Author's response:** In general, the choosing of two models and various combinations of physical schemes is motivated by providing a very robust estimation of investigated changes. Considering the fact that we are interested mainly in local urban induced changes, we tested nearly all available schemes of urban canopies, boundary layer (connected with schemes of surface layer) and convection (for WRF model), and schemes of boundary layer, convection and microphysics (for RegCM). Convection parameterization can have potential effects on vertical transport of heat and moisture from the urban boundary layer while the model treatment of microphysics marks the hydrological budget over cities that influences the precipitation, latent heat release etc. The selection of different combination of model schemes is further based on their availability and restrictions in their different combinations.

**Referee's Comment #3:** Regarding the presentation of the urban results in section 3.2, a selection of those models who performed best in the overall evaluation would make much more sense.

**Author's response:** The general evaluation of model results in section 3.1 describes overall biases of temperature and precipitation, extended by comparison with station data for other variables. However, it does not serve as a validation of any urban-induced change, thus it does not tell us much about the accuracy of such changes. Moreover, it is not easy to say, which model is the best, from an overall view. E.g. simulations with the BEP+BEM urban model make higher biases in winter max/min temperature, but on the other hand, the wind speed is closer to reality (Fig. 4 and 5). Our study intends to provide a physical ensemble of the different components of UMI therefore, we present

urban-induced changes of all simulations and/or averaged ones.

**Referee's Comment #4:** What aspects of the study have been surprising for the authors?

**Author's response:** Over the fact that the horizontal resolution is still relatively coarse (9 km), urban-induced changes (not only UHI) are clearly visible and mostly statistically significant. Especially the urban alteration of cloud cover and sub-grid scale precipitation in summer (Fig. 11 and 12), which are also described in observation-based studies. Further, the great influence of microphysics scheme (namely Nogherotto scheme in RegCM) on overall temperature and precipitation.

**Referee's Comment #5:** Can the authors also give a recommendation for the combination of physical packages using either WRF or RegCM?

**Author's response:** For our experiments with WRF that follow this study, we used the WU1L82C5 combination, e.i. SLUCM urban model, BouLac PBL and Eta SFC scheme, together with Grell-3D convection, as a less computational demanding compromise to BEP+BEM urban model, which needs more model levels. In terms of the RegCM model, probably the RUS simulation (UW PBL scheme, Tiedtke convection and SUBBEX microphysics scheme) makes the lowest biases in our domain.

---

## Author Comment (AC2) · 7 Oct 2020

We would like to thank to Anonymous Referee #1 for all comments, suggestions and corrections in his review of our manuscript. They cover issues which were resolved in the initial review, therefore they have been already taken into account by the authors and incorporated into the text. Nevertheless, we provide our point-by-point responses here too:

**Referee's Comment #1:** Abstract: the abstract should mention that you only study European cities. In terms of morphology they are substantially different North American cities that this caveat makes sense to mention.

Author's response: Corrected.

**Referee's Comment #2:** Abstract: you indicate that there is a substantial sensitivity of the model to the selected PBL schemes and urban canopy scheme. It would be more attractive for the reader if you can add a recommendation which settings are preferred.

**Author's response:** The primary task of the paper was not to provide an optimal model setup to describe urban climate but rather to examine the sensitivity of the modelled urban climate and its contrast with urban vicinity to different models and model configurations. Nevertheless, we added a such 'recommendation' to the discussion and conclusion based on Fig. 3, 4 and 5.

Referee's Comment #3: Ln 12: choice => selection

Author's response: Corrected.

Referee's Comment #4: Ln 13: hyphenation: boundary-layer scheme

Author's response: Corrected.

Referee's Comment #5: Ln 31: hyphenation: boundary-layer structure

Author's response: Corrected.

**Referee's Comment #6:** Ln 59: hyphenation: boundary-layer turbulence. Please check the manuscript throughout.

Author's response: Corrected everywhere.

**Referee's Comment #7:** Ln 76: model resolution of 9 km: defend why this is sufficient to represent cities sufficiently. The resolved scales will be 5 \* 9 km = 45 km, which means only cities ofthat scale are appropriately resolved, but there are not that many cities in Europe ofthat scale. I the feeling the cities on Belgium and Hungary are larger in your Fig. 1 than in reality.

Author's response: As mentioned in the manuscript, the urban land-use is differently represented in WRF and RegCM: in RegCM it is defined as fractional landuse which

**ACPD**
allows even small urban areas to be resolved as fractions of the 9km×9km grid-box. In WRF, grid-box is considered 'urban', when the urban land-use has the highest fraction, i.e. it does not have to be strictly over 50 % of grid-box area. Moreover, urban land-cover can be increased by small towns and villages located in the grid-box near the city, which seemingly increase the city size. However, big cities such as Berlin, Prague, Warsaw or Budapest, analysed in this paper, are covering multiple model grid-boxes themselves and thus are represented sufficiently even at 9km×9km resolution.

**Referee's Comment #8:** Ln 78: 2015–2016: please defend why these years have been selected. 2015 is a rather warm year in Europe, so how representative is the selected period.

**Author's response:** Indeed, 2015 is a warmer year than average (not so for 2016) (https://climate.copernicus.eu/european-temperature). However, years 2015 and 2016 are comparable to other years in the last decade (e.g. 2014, 2018) for Europe. Moreover, the variability of the local climate characteristic for individual cities chosen in the study is certainly larger (e.g. Hamburg with about 8° C annual average temperature versus Beograd 12° C; source: climate-data.org) than the year to year variability of the average European climate. We thus conclude that the spread of the magnitude of the urban meteorological effects and the 'urban meteorology island' given by considering a wide range of cities is well above the spread given by choosing different years during recent decade.

**Referee's Comment #9:** Ln 85: please add some sentences that defend why you have selected these schemes. I expect you have not selected them randomly but that there was a certain strategy or you built upon earlier studies.

**Author's response:** The selection is based on the availability of the schemes and the restrictions for their different combinations as well as on the expected impact on the 'urban meteorology island'. For WRF: BEP+BEM urban model works only with MYJ and Boulac PBL schemes combined with the Noah land-surface model. The MYJ PBL
scheme works only with Eta surface-layer scheme (see e.g. WRF-ARW user's guide). Further, schemes usable in WRF-Chem model with indirect aerosol effect are tested (Purdue Lin microphysics scheme) and those that enable cumulus radiation feedback (listed convective schemes). These simulations are planned to be used in a following study with the radiative feedback of the emissions from the selected cities. The RRTMG radiation scheme was chosen as an efficient radiative transfer model that is fast yet has sufficient complexity. For RegCM: two PBL schemes are available (Holtslag and UW) so these were altered as the PBL model description greatly influences the meteorology in the urban canopy. Further, the two most widely used convective schemes were tested, the Grell and Tiedtke schemes with the consideration that convection is an important process that removes heat and moisture from urban areas influencing urban climate. Finally, three microphysical schemes are available in RegCM (for the used version 4.7) and these were combined with the PBL and cumulus schemes. The urban canopy model could not be altered as only the CLM4.5/CLMU was available for RegCM simulations.

**Referee's Comment #10:** Table 1: elaborate the table caption. The caption should be placed above the table. Idem for Table 2 and 4.

Author's response: Changed for all tables.

**Referee's Comment #11:** Table 1: please add a sentence that elaborates on the experiment abbreviations.E.g. the "E1U1L82C5" is not naturally related to the <SLUCM, 40, BouLac, Eta, Grell-3D> experiment. All experiment abbreviations start with "E1" so E1 can be removed. Idem for Table 2

**Author's response:** E1 is removed from all abbreviations, added 'W' on the beginning in Table 1, denoting WRF model, similarly to RegCM model in Table 2. Changed in all tables and figures, including the text.

**Referee's Comment #12:** Ln 118: simulations with MYJ PBL and Eta SFL schemes give notable underesti-mations (up to 2 deg C): link to literature, this MYJ behaviour is

ACPD
well known.

**Author's response:** This feature is commented and referenced (by Zhong et al., 2017) in the first paragraph of the Discussion section.

**Referee's Comment #13:** Figure 3: I have reservations against figure 3. Not about the contents but about the plot type. Now the results are shown as time series or at least the lines connect the different experiments. However there are no links between the connected experiments.So the results should be presented differently, e.g. as bar graphs. Idem for Fig 6.

**Author's response:** X-axis does not mean time axis automatically, lines between points are removed from the plot, points are enlarged (Fig. 3 and 6).

**Referee's Comment #14:** Ln 124: Daily ECAD values from selected stations are used for more detailed model validation over european urban areas, which are the main focus of the paper:European should be capitalized.

Author's response: Remark accepted.

**Referee's Comment #15:** Figure 4: please label all figure a,b,c, etc. This is much more easy for referencing

**Author's response:** We believe that subplots are sufficiently denoted by variable names and statistical quantities, but in any case, the publisher places "a)", "b)" etc. subplot indicators, automatically during the typesetting process (including placement in the figures caption).

**Referee's Comment #16:** Figure 4: panel T2max and SWDOWN should have an y axis that is better adjusted(less wide range) to the data.

Author's response: Comment accepted. Figure 4 modified.

**Referee's Comment #17:** Figure 4: in columns 2 and 3 it is very difficult to see what are the differences between the runs. Better to start the y axis at a much higher value.
Idem for Fig 5.

Author's response: Comment accepted. Figures 4 and 5 are modified.

**Referee's Comment #18:** Figure 4 and 5: if these statistics are averaged over all cities in Table 3, it remains unclear how they are influenced by certain sites or not.

**Author's response:** We are unsure if we understand the reviewer's comment properly. We tried to avoid a city-by-city validation to present rather the all-city-averages, which gives an indication of how the models are able to capture the urban climate for cities from an entire region. It is clear that models are more successful for some cities and less for others, but our aim is not to present individual city statistics.

**Referee's Comment #19:** Fig 8 and 9: the header "surface heat island" is misleading since this is not plotted according to the caption. Furthermore I do not understand what is the functionality of these plots if you only show satellite data.

**Author's response:** The header of Fig. 8 and 9 is "Surface urban heat island" as we plotted the surface temperatures (or skin-temperatures). The goal of presenting these two figures was to show the urban impact on surface temperature seen by remote sensing methods (satellites in this case). SUHI is a major component of what we call 'urban meteorological island' in our paper and we therefore decided to also show some observational based evidence of this component. Satellite measurements have the great advantage of showing the spatial distribution of the measured quantity in contrast with e.g. station based data.

**Referee's Comment #20:** Ln 198: please be more precise here: about which temperature are writing here. It is daily mean T2m?

**Author's response:** Yes, they are daily means of T2m during summer and winter season. The information has been added into the text.

**Referee's Comment #21:** Ln 202: absolute humidity is the density of the vapour pressure so unit should be g/m3
**Author's response:** Following the Meteorological Glossary of American Meteorological Society, we changed it to "specific humidity", which describes mixing ratio, independent to adiabatic expansion or compression, and can be measured in kg kg-1 (i.e. it is dimensionless).

**Referee's Comment #22:** Ln 333: it is unclear whether the statement of BEP+BEM is an advertisement for this scheme or not. Is this scheme the best, despite the biases you report about?

**Author's response:** It is not easy to say whether the BEP+BEM is better or not – it produces greater biases in temperature but smaller biases in wind speed in comparison to other schemes.

**Referee's Comment #23:** Ln 349: I have the feeling the authors are somehow too positive about the satellite data. As far as I understand them, they can only be applied for cloud free days, and this does not occur very often, so they may give a biased picture. Please comment.

**Author's response:** Yes, surface temperature is measured only during clear-sky days by satellites, but over this fact, it still gives useful insight in surface temperature distribution around large cities, thus demonstrating the main component of the 'urban meteorology island' using remote sensing based data, which enables observation over larger areas rather than point measurements. The information about clear-sky days is added to the text.

ACPD